# A Novel Mutation Associated with Neonatal Lethal Cardiomyopathy Leads to an Alternative Transcript Expression in the X-Linked Complex I *NDUFB11* Gene

**DOI:** 10.3390/ijms24021743

**Published:** 2023-01-16

**Authors:** Guillermo Amate-García, María Juliana Ballesta-Martínez, Pablo Serrano-Lorenzo, Rocío Garrido-Moraga, Adrián González-Quintana, Alberto Blázquez, Juan C. Rubio, Inés García-Consuegra, Joaquín Arenas, Cristina Ugalde, María Morán, Encarnación Guillén-Navarro, Miguel A. Martín

**Affiliations:** 1Grupo de Enfermedades Mitocondriales y Neuromusculares, Instituto de Investigación Hospital 12 de Octubre (imas12), 28041 Madrid, Spain; 2Centro de Investigación Biomédica en Red de Enfermedades Raras (CIBERER), 28029 Madrid, Spain; 3Sección de Genética Médica, Servicio de Pediatría, Hospital Clínico Universitario Virgen de la Arrixaca, Instituto Murciano de Investigación Biosanitaria (IMIB) Pascual Parrilla, 30120 Murcia, Spain; 4Facultad de Medicina, Universidad de Murcia, 30120 Murcia, Spain; 5Servicio de Genética, Hospital Universitario 12 de Octubre, 28041 Madrid, Spain

**Keywords:** *NDUFB11* gene, mitochondrial complex I, neonatal cardiomyopathy, splicing, heart muscle, skeletal muscle

## Abstract

We report a neonatal patient with hypertrophic cardiomyopathy (HCM), lactic acidosis and isolated complex I deficiency. Using a customized next-generation sequencing panel, we identified a novel hemizygous variant c.338G>A in the X-linked *NDUFB11* gene that encodes the NADH: ubiquinone oxidoreductase subunit B11 of the mitochondrial respiratory chain (MRC) complex I (CI). Molecular and functional assays performed in the proband’s target tissues—skeletal and heart muscle—showed biochemical disturbances of the MRC, suggesting a pathogenic role for this variant. In silico analyses initially predicted an amino acid missense change p.(Arg113Lys) in the NDUFB11 CI subunit. However, we showed that the molecular effect of the c.338G>A variant, which is located at the last nucleotide of exon 2 of the *NDUFB11* gene in the canonical ‘short’ transcript (sized 462 bp), instead causes a splicing defect triggering the up-regulation of the expression of an alternative ‘long’ transcript (sized 492 bp) that can also be detected in the control individuals. Our results support the hypothesis that the canonical ‘short’ transcript is required for the proper NDUFB11 protein synthesis, which is essential for optimal CI assembly and activity, whereas the longer alternative transcript seems to represent a non-functional, unprocessed splicing intermediate. Our results highlight the importance of characterizing the molecular effect of new variants in the affected patient’s tissues to demonstrate their pathogenicity and association with the clinical phenotypes.

## 1. Introduction

Mitochondrial complex I (CI) or NADH: ubiquinone reductase H(+)-translocating-(EC 7.1.1.2) is the largest multimeric enzyme complex of the oxidative phosphorylation (OXPHOS) system. CI consists of 44 different structural subunits (37 nuclear and 7 mitochondrial-encoded). At least 15 nuclear assembly factors are known to be involved in CI holoenzyme biogenesis [1,2,3]. CI is a modular L-shaped enzyme structurally formed by: (i) a hydrophilic peripheral arm projected to the mitochondrial matrix comprising two functional modules, the N-module (NADH dehydrogenase) and the Q-module (ubiquinone reduction); and (ii) a hydrophobic membrane arm integrated into a mitochondrial inner membrane (MIM) formed by the P-module (proton pumping) [4,5,6]. CI represents the entry point of electrons from NADH into the mitochondrial respiratory chain (MRC), which travel across complexes CI–CIII–CIV and drive proton pumping toward the intermembrane space to generate a mitochondrial membrane potential used for adenosine triphosphate (ATP) synthesis using the H^+^-ATP (CV) of the OXPHOS system [7]. In addition, CI is the main site of generation of reactive oxygen species (ROS), which can modulate a large number of signaling pathways [8,9].

Isolated CI deficiency (MIM #252010) is the most frequent cause of OXPHOS enzymatic deficiencies (≈30%). Genetically, it is caused by pathogenic variants in CI structural subunits and assembly factors, which mostly lead to early onset autosomal recessive or maternal inherited disorders displaying a wide phenotype spectrum that either involve several organs or only specific tissues [7,8]. The organs most frequently affected are the central nervous system, skeletal muscle, myocardium, liver and kidney, and typically are clinically manifested as Leigh syndrome, neonatal cardiomyopathy, fatal infant lactic acidosis (FILA), Leber hereditary optic neuropathy (LHON), hepatopathy with tubulopathy or mitochondrial encephalopathy with lactic acidosis and stroke-like episodes (MELAS) [10,11].

Only two CI structural subunits genes, *NDUFA1* and *NDUFB11*, are located on the X chromosome [12,13,14,15]. The *NDUFB11* gene encodes the CI accessory subunit, NADH: Ubiquinone Oxidoreductase Subunit B11 that is located at the distal membrane arm of the P-module (ND4 or P_D-a_). Thus far, several pathogenic de novo and X-linked inherited variants have been reported in 15 patients presenting with hypertrophic or histiocytoid cardiomyopathy and sideroblastic anemia, accompanied by other clinical manifestations such as failure to thrive, myopathy, short stature, epilepsy or microphthalmia with linear skin defects (MLS syndrome) [15,16,17].

In this study, we identified and characterized a novel splicing variant in the *NDUFB11* gene reported in a neonatal male patient with fatal hypertrophic cardiomyopathy (HCM), lactic acidosis and single CI deficiency. We also characterized the biochemical and molecular effects of this variant in the patient’s heart and muscle, and showed the possible protective effect of a skewed X-chromosome inactivation in the proband’s family, thus clarifying the diverse functionality of the two expressed *NDUFB11* transcripts.

## 2. Results

### 2.1. Biochemical, Genetic and Molecular Characterization of the NDUFB11:c.338G>A Variant

The proband’s skeletal muscle homogenates exhibited an isolated CI deficiency (22.7% with respect to the 2.5th percentile of controls) (Table 1).

The proband’s muscle whole mtDNA deep-sequencing failed to detect any potential pathogenic variant and predicted that the patient belonged to the H1a mtDNA haplogroup.

A customized next-generation sequencing (NGS-OXPHOS) panel including structural and assembly factors of CI (Appendix A) identified 174 variants in the proband’s muscle DNA, but none of them were categorized in the ClinVar or InterVar tools as pathogenic. Variant filtering by discarding sequencing errors and ‘deep intronic’ variants, MAF < 0.01 in population databases, resulted in the identification of two variants. One variant, a hemizygous c.338G>A was located in the last nucleotide of the exon 2 of the X-linked CI subunit gene *NDUFB11,* predicting an apparent missense change in the protein p.(Arg113Lys). The variant was absent in population databases, such as the 1000 Genomes Project, gnomAD or exome variant server and was categorized as deleterious by several in silico predictors such as LRT, MutationTaster, CADDPhred, GERP, PhastCons and phyloP.

Prediction algorithms for mRNA splicing, including MutationTaster, Human Splicing Finder, dbNSFP and dPSI suggested that the c.338G>A change could affect the 5′ splicing donor site of the *NDUFB11* intron 2. Following the American College of Medical Genetics and Genomics and the Association for Molecular Pathology (ACMG/AMP) guidelines for the interpretation of sequence variants [18], the c.338G>A variant was classified as likely pathogenic.

Sanger sequencing confirmed the presence of the hemizygous c.338G>A variant in the proband’s blood and skeletal muscle. Family segregation analyses in blood DNA showed the presence of the variant in heterozygosis in the proband’s mother and a maternal aunt, whereas it was absent in the father (Figure 1a,b).

### 2.2. cDNA Analysis of NDUFB11

Proper splicing of *NDUFB11* mRNA could be altered by the c.338G>A variant since it is located in the last nucleotide of exon 2 of the ‘canonical’ transcript (NM_001135998), which has a size of 462 bp. To study the potential alteration in the splicing of *NDUFB11* mRNA, we retrotranscribed RNA to cDNA from two tissues, skeletal muscle and heart, from both the proband and control individuals. Analysis of the PCR-amplified *NDUFB11* cDNA and subsequent sequencing of the electrophoretic bands revealed, both in the proband and control tissues, the presence of a longer transcript of 492 bp that included a retention of an additional 30 bp with regards to the ‘canonical short’ transcript. This transcript was previously described as an ‘alternative’ transcript of *NDUFB11* (NM_019056). The levels of the canonical short cDNA analyzed using quantitative PCR (qPCR) were undetectable both in the patient’s skeletal muscle and heart tissues, whereas the levels of the alternative long cDNA were up-regulated in the proband’s tissues with respect to the controls (114% increase in skeletal muscle and 28% in heart tissue) (Figure 2).

### 2.3. X Chromosome Inactivation Pattern by HUMARA Assay

In order to investigate the X chromosome inactivation (XCI) pattern in the proband’s mother and maternal aunt (II-2, Figure 1), we performed a methylation status analysis using *HUMARA* assay [20]. This study showed a skewed X inactivation pattern with a preferentially inactivated allele ratio of 78%:22% in the mother and 80%:20% in the tested maternal aunt (Appendix A). An assay control test was also performed in four non-carrier females that showed a mean XCI ratio of 49%:51% (Appendix A). Likewise, a control male was tested (Appendix A) showing a complete HpaII digestion of the unmethylated (active) X chromosome, ensuring a precise XCI patterns estimation. The amplification and alignment of the peaks obtained in the fragment analysis of the non-digested proband (male) allele revealed that his X allele carrying the *NDUFB11* hemizygous mutation corresponded with the preferentially inactivated X allele observed both in the mother and maternal aunt (Appendix A). Moreover, there were no XCI differences between the asymptomatic mother and the maternal aunt diagnosed with incidental HCM.

### 2.4. Western Blot of NDUFB11 and Subunits of MRC Complexes in Proband’s Tissues

Western blot analysis of the proband’s muscle showed very low levels of the NDUFB11 protein, which was undetectable in the heart tissue, compared with the controls. The analysis of proteins belonging to the different CI modules showed similar patterns for NDUFA9 (Q module) and NDUFB8 (P module), the latter undetectable in both tissues, as well as reduced levels of NDUFV1 (N module) in both tissues. Immunodetection of representative subunits from other MCR complexes, such as SDHA (CII), Core2 (CIII) and COXII (CIV) displayed no differences in band intensities compared to the controls (Figure 3a).

### 2.5. Steady-State Levels of Mitochondrial Supercomplexes and CI In-Gel Activity (CI-IGA)

First dimension Blue-Native PAGE (1D-BNE) either followed by complex I in-gel activity (CI-IGA) assay or by Western blot using antibodies targeting CI subunits NDUFA9 and NDUFB11 revealed a specific severe reduction of CI assembled into supercomplexes (both in SC I + III_2_ + IV_1–2_ or “respirasome” as well as in SC I + III_2_) in the proband’s skeletal muscle and heart tissue, compared with aged-matched controls (Figure 3a). Incubation with Core2 (CIII subunit) and COX5a (CIV subunit) antibodies confirmed reduced amounts of respirasome and SC I + III_2_, along with the accumulation of SC III_2_ + IV_1_ and dimeric CIV (IV_2_) in the proband’s heart.

The second dimension of BNE (2D-BN/SDS-PAGE) performed in heart tissue confirmed the alterations observed in 1D-BNE, showing lower levels of CI, CIII and CIV in the “respirasome”, a reduction of SC I + III_2_ and the accumulation of SC III_2_ + IV_1_ and CIV_2_ in the proband (Figure 3b).

## 3. Discussion

Mammalian complex I (NADH: Ubiquinone oxidoreductase) is a huge multimeric enzyme structure of approximately 1 MDa. It is composed of an assembly of 45 structural subunits regulated by more than 15 assembly factors, of which seven subunits are encoded by mtDNA and the remaining are encoded by genes located in nuclear chromosomes; it constitutes the electron’s entry point of NADH oxidation into the mitochondrial respiratory chain (MRC) [2,4,5]. Isolated CI deficiency represents the main cause of OXPHOS disorders (about 25–30% of mitochondrial diseases) associated with a wide genetic–phenotypic spectrum leading to multiorgan or tissue-specific disorders [8,21]. The most common clinical presentations are Leber’s hereditary optic neuropathy (LHON), Leigh syndrome, leukoencephalopathy, neonatal cardiomyopathy, fatal neonatal lactic acidosis (FILA), childhood-onset mitochondrial encephalomyopathy and lactic acidosis and stroke-like episodes syndrome (MELAS) [21]. In patients with CI deficiency, mutations have been found in all of the core subunits (catalytic function), in half of the accessory subunits (15/30) and in most of the assembly factors (12/15), which are mainly associated with autosomal recessive inheritance in pediatric cases, and in minor proportion also in adult individuals (>20 years, principally caused by mtDNA mutations) [10,11,22].

In this study, we describe a newborn male who presented with HCM and lactic acidosis, developed coagulopathy and multiorgan failure and died at 48 h of life. The skeletal muscle biopsy showed a single CI defect with no histomorphological abnormalities. After discarding mutation in the mtDNA, we used a customized NGS gene panel to identify a novel hemizygous variant c.338G>A; p.(Arg113Lys) in the *NDUFB11* gene. This variant was located at 3′ end of exon 2, and several predictors suggested it could affect splicing. This variant was classified as likely pathogenic according to ACMG/AMP guidelines [18]. To assess the pathogenicity of this variant, we performed a functional validation in skeletal and heart muscle from the patient, using an analysis of transcripts and protein expression as well as studies of OXPHOS SCs assembly.

The analyses of cDNA revealed that the variant alters the canonical donor site at the 3′-end of exon 2 and results in an alternate transcript with an additional 30 bp (NM_019056). At physiological status, the *NDUFB11* gene generates two mRNA transcript isoforms: (i) an abundant short transcript of 462 bp (NM_001135998) coding for a 153 aa protein, and (ii) a less-expressed longer transcript of 492 bp (NM_019056), whose biological role is still not well understood [23]. This long transcript is predicted to encode for a 163 aa protein as a result of an alternative splicing between exons 2–3 that would add 30 bp to the exon 2 boundary. While there is some debate about which of these two *NDUFB11* mRNAs represents the canonical and the alternative transcript, several reports considered the shorter transcript (NM_001135998) as the main canonical transcript [23,24,25]. For instance, Petruzzella et al. hypothesized that the 10 amino acids insertion present in the long isoform would lead to a severe alteration in the secondary structure of NDUFB11, suggesting a possible pathological role for this specific transcript [25]. In this work, we showed using reverse RT-qPCR that c.338G>A alters splicing, leading to the absence of expression of the canonical short transcript in the proband’s skeletal muscle and heart tissue, thereby confirming its pathogenic nature. The c.338G>A nucleotide change would impede the splicing machinery to recognize the canonical 5′donor-site leading to the canonical ‘short’ transcript, and as a result, an alternative, non-functional, not fully processed transcript was the only one expressed in large amounts in the proband’s analyzed tissues. Despite the high expression levels of the alternative long transcript, only scarce levels of NDUFB11 were detected in the skeletal muscle and no protein was detected in the heart (the primarily affected tissue according to HCM). These results indicated that the alternative transcript failed to synthetize a stable protein or only produced minimal amounts of NDUFB11, whereas the canonical transcript would be required for appropriate NDUFB11 synthesis.

As the c.338G>A variant disturbed normal splicing of *NDUFB11* mRNA, we also analyzed the pathophysiological consequences on NDUFB11 protein levels and on the overall assembly state of the MRC complexes and SCs. CI is assembled from 14 “core” subunits with catalytic functions (ND1-ND6, ND4L, NDUFV1-V2, NDUFS1, S2, S3, S7 and S8) and 30 accessory subunits. Although most of the function of the supernumerary or accessory subunits are poorly understood, the pathogenic variants found in some of these proteins in patients with several OXPHOS disorders reveal that these subunits are required for CI stability and assembly [26]. Complexome profiling studies using mtDNA translation inhibitors previously revealed NDUFB11 as one of the first subunits involved in the initial steps of assembly of the CI membrane arm [5], a step essential to proceed with the formation of subsequent CI intermediates assembled at later stages of CI assembly. In agreement, loss of the NDUFB11 protein led to overall decreased levels of all CI structural modules, suggesting a severe CI assembly defect, especially in the proband’s heart. This explains why the proband’s skeletal muscle and heart tissues showed low levels of the remainder CI subunits, and supports the essential role of NDUFB11 in CI biogenesis [27,28,29]. In turn, decreased CI steady-state levels led to a remarkable reduction of SCs I + III_2_ + IV_1-2_ and I + III_2_ and to the accumulation of SC III_2_ + IV_1_ and dimeric CIV (CIV_2_) in the proband’s heart. These results are consistent with those previously observed in other cellular models of CI deficiency [30,31]. CI in-gel activity (CI-IGA) confirms a remarkable reduction of isolated CI and SCs activity in the proband’s heart compared to the control, which is consistent with the decreased CI activity observed by spectrophotometry in the proband’s skeletal muscle.

*NDUFB11* is a dominant X-linked nuclear gene (Xp11.23) encoding for a small accessory CI structural subunit of about 17.3 kDa [15]. To date, 8 de novo and maternally inherited pathogenic *NDUFB11* variants have been detected in 15 patients manifesting with a wide clinical spectrum. Cardiomyopathy (67% of reported patients) and sideroblastic anemia (47%) were the primary phenotypes, together with other less frequent clinical manifestations such as failure to thrive (13%), myopathy (13%), short stature (33%), epilepsy (13%) and microphthalmia with linear skin defects (MLS syndrome) (13%) [16,17,28]. CI deficiencies usually show a very poor prognosis and about 80% of patients die before the age of 10 [10]. Our patient died 48 h after birth with HCM. An abortion and two perinatal deaths were also reported in the proband’s cousins (from a maternal aunt, II-2). Dilated cardiomyopathy was well documented in the family, despite most *NDUFB11* mutations displaying variable clinical phenotypes somewhat different from other CI defects. Of the 15 previously reported *NDUFB11* patients, only 2 were associated with early death: a 6-month female patient having MLS syndrome with histiocytoid cardiomyopathy [24] and a male who died at 55 h after birth due to a lethal infantile mitochondrial disease with cardiac failure [29]. Furthermore, Van Rahden et al. described an abortion (XX) at 24 weeks of gestation due to cardiac alterations [24]. X-linked disorders are characterized by more severe clinical symptomatology in male subjects and a higher number of male patients. This phenomenon can be explained by the fact that males, compared to females, only receive a single copy of the X chromosome. In the other hand, lyonization or XCI is a random stochastic phenomenon that takes place in the embryonic first stages in females to compensate for double X chromosome genetic dosage. Nevertheless, it has been described as preferential XCI (skewed XCI) due to the pathogenic variants in X-linked diseases in healthy carrier females [20,32]. The influence of completed or skewed XCI could explain why female heterozygous carriers of pathogenic variants in the *NDUFB11* gene are either asymptomatic or show different susceptibility to express phenotypes with respect to very severely affected males harboring hemizygous pathogenic variants. A protective effect of skewed XCI pattern due to pathogenic variants in the *NDUFB11* gene has been previously suggested in two healthy mothers of three females with an X-linked dominant MLS syndrome with histiocytoid cardiomyopathy, epilepsy, developmental delay, failure to thrive and hypotony [24], as well as in a healthy mother and sister of a 23-year-old male presenting with sideroblastic anemia, myopathy and lactic acidosis [17]. The XCI analysis in our family revealed a skewed 78%:22% X inactivation ratio in the carrier proband’s mother and 80%:20% in a carrier maternal aunt for the mutated allele, suggesting a protective effect of XCI in these carriers against the pathogenic variant that could partially explain their asymptomatic phenotype. However, these results fail to explain the HCM found in the maternal aunt compared to the healthy mother. A less severe, incidental HCM phenotype detected at the age of 47 in a cardiac study previous to myoma uterine surgery, or even a different etiology, could explain this event in the maternal aunt, who had two children with dilated cardiomyopathy (DCM) that died within months after birth.

Furthermore, loss of function variants (p.Trp85*, p.Arg88*, p.Tyr108* and p.Arg134Serfs*3) were the type of variant present in all previously reported symptomatic female probands in contrast with missense variants harbored by severe (p.Glu121Lys) or mildly affected (p.Ser96Pro, p.Pro110Ser) males, or with those males carrying an in-frame variant (p.Phe93del) associated with sideroblastic anemia [28]. In this context, interestingly, our early-death male proband harbors a splicing loss of function variant.

All of this evidence suggests that the severity of the phenotype manifested in patients having a pathogenic variant in the *NDUFB11* gene is presumably underlaid by a complex interplay between the type of variant, the extent of dysfunctional OXPHOS as a result of complex I deficiency, and in females also by the presence of skewed XCI.

In summary, we showed that a novel exonic c.338G>A variant in the nuclear *NDUFB11* gene, which was predicted to be a missense variant is actually a splicing variant and is associated with mitochondrial X-linked isolated CI deficiency and neonatal lethal hypertrophic cardiomyopathy. Moreover, our results suggest the canonical short transcript (NM_001135998) is required for the correct NDUFB11 protein synthesis, as well as for CI assembly and activity, while the alternative long *NDUFB11* transcript seems to have no functional role. However, further work is warranted to investigate the physiological role of this transcript. In addition, we showed the pathogenic effect of the *NDUFB11* variant in two tissues in the target patient, skeletal and heart muscle, including their effects on the steady-state levels of the mitochondrial supercomplexes (SCs).

## 4. Materials and Methods

### 4.1. Patient and Family

The study was approved by the Ethics Committee of the ‘Hospital Universitario 12 de Octubre’ (Madrid, Spain) and was performed in accordance with the Declaration of Helsinki for Human Research. Written consent was obtained from the analyzed patients or proband’s parents.

The family pedigree is depicted in Figure 1a. The proband (III-3) was the first child of non-consanguineous asymptomatic parents. He presented at birth with lactic acidosis and dysmorphic features. A postnatal echocardiogram displayed a septum alteration revealing hypertrophic cardiomyopathy (HCM). At 24 h, the child showed an acute clinical deterioration presenting with respiratory distress and pallor skin and was admitted to the neonatal intensive care unit. Laboratory data showed hyperlactatemia (20.0 mM, normal < 2.0), with normal plasma glucose and ammonia. He died at 48 h of age from coagulopathy and multiorgan failure. His maternal aunt (II-2) who was diagnosed incidentally of HCM at 47 years of age in a routine cardiac study previous to a myoma uterine surgery, had two male descendants with dilated cardiomyopathy (DCM), who died in the perinatal period (III-1) and at 3 months of age (III-2). Two brother and three sisters of the proband’s mother referred no cardiac disturbances. One of the mother’s sisters (II-5) had a voluntary interrupted pregnancy due to a malformed fetus with intrauterine fetal death. The proband’s grandparents had no cardiac abnormalities.

### 4.2. Mitochondrial Respiratory Chain Analysis

MRC complexes and citrate synthase enzyme activities in skeletal muscle were determined spectrophotometrically as previously described [33,34].

### 4.3. Molecular Genetic Studies

#### 4.3.1. Whole Mitochondrial Genome Next-Generation Sequencing

Whole mitochondrial genome deep sequencing (mtDNA-WGS) was performed using long-range PCR amplification of the entire mtDNA and following the protocols based on Ion-PGM sequencer (Life Technologies, Carlsbad, CA, USA). Bioinformatic analysis, and variant calling, annotation and prioritization was carried out using pipeline integrating variant calling format (VCF) files with Mitomap database and MitImpact predictors using in-house scripts as reported [34]. The mtDNA haplogroup was predicted using the Haplogrep2 platform [35].

#### 4.3.2. Next-Generation Sequencing (NGS)—OXPHOS Panel

The nuclear encoded structural subunits and assembly factors of the OXPHOS system were analyzed using a customized “NGS-OXPHOS” panel, which allow the sequencing of 133 genes (Appendix A). This panel was designed with an Ampliseq^TM^ designer v1.2 (Life Technologies Carlsbad, CA, USA) and sequencing was performed in Ion-PGM Torrent platform (LifeTechonologies, Carlsbad, CA, USA) with a mean read depth of 100X. Sequence alignment (ref.GRCh37/hg19) and variant detection was performed in Torrent Suite (TMAP-variantCaller plugin, Thermo Fisher Scientific,, Waltham, MA, USA). Variant annotation and prioritization were conducted through integration of in-house scripts with Annovar [36]. Variant prioritization was based assuming an autosomal recessive or X-linked inheritance as follows: (1) status of the variants in the ClinVar database using the InterVar tool; (2) minor allele frequency (MAF) < 0.01 in population databases such as the Genome Aggregation Database (gnomAD, https://gnomad.broadinstitute.org) and the 1000 Genomes Project database (http://browser.1000genomes.org); (3) variant pathogenicity predictors including SIFT (http://sift.jcvi.org), PolyPhen-2 (http://genetics.bwh.harvard.edu/pph2), LRT (http://www.genetics.wustl.edu/jflab/lrt_ query.html), MutationTaster (http://www.mutationtaster.org), M-CAP (http://bejerano.stanford.edu/mcap/), PROVEAN (http://provean.jcvi.org/index.php), and CADD Phred (http://cadd.gs.washington.edu) and splicing predictors such as dbNSFP (https://sites.google.com/site/jpopgen/dbNSFP), dPSI (difference in percentage spliced in) and Human Splicing Finder (http://www.umd.be/HSF3/); (4) assessment of phylogenetic conservation using Genomic Evolutionary Rate Profiling (GERP) and the Phylogenetic Analysis with Space/Time models (PHAST) programs: phastCons and phyloP. All databases and predictors URLs were accessed on 15 October 2019.

#### 4.3.3. Sanger Sequencing of the *NDUFB11* Gene

Genomic DNA (gDNA) was isolated from total blood, skeletal muscle and heart tissue using QIAamp DNA Mini Kit (QIAGEN, Hilden, Germany) following the manufacturer’s recommendations. Sanger sequencing was performed in the proband and his parents and a maternal aunt (II-2 in Figure 1a) to confirm the presence and segregation of the novel variant identified in the *NDUFB11* gene. Polymerase chain reaction (PCR) amplification of exon 2 of *NDUFB11* gene (Appendix A) was performed in an Applied Veriti Thermal Cycler (Applied Biosystems, Waltham, MA, USA). PCR products were purified using Ilustra GFX PCR DNA and Gel Band Purification Kit (GE-Healthcare, Amersham, UK). After purification, PCR product was sequenced using BigDye X Terminator (Applied Biosystems, Waltham, MA, USA) and signal was detected in an Applied Biosystem 3130xl Genetic Analyzer (Applied Biosystems, Waltham, MA, USA).

#### 4.3.4. mRNA Analysis of *NDUFB11*

RNA isolation from skeletal muscle and heart tissue was performed using TRIzol Reagent (Life Technologies, Carlsbad, CA, USA). Two µg RNA were retrotranscribed with SuperScript IV (Invitrogen, Waltham, MA, USA) and *NDUFB11* cDNA was amplified by PCRs using specifically designed primers (Appendix A). PCR products were visualized in 2.5% agarose gel and were purified using the Illustra GFX PCR DNA and Gel Band Purification Kit (GE-Healthcare, Amersham, UK). cDNA was sequenced in a 3130xl Genetic Analyzer (Applied Biosystems, Waltham, MA, USA). The relative levels of the canonical (NM_001135998) and alternative (NM_019056) *NDUFB11* mRNAs in skeletal muscle and heart of proband were analyzed with the FastStart Essential DNA Green Master kit (Roche Diagnostics, Mannheim, Germany) on LightCycler 96 (Roche Diagnostics, Mannheim, Germany) as described by manufacturer´s protocol using specific primers designed with PrimerQuest (Integrated DNA Technologies, Coralville, IA, USA) (Appendix A). Hipoxantine-guanine phosphoribosiltransferase 1 (*HPRT1*) was used as a constitutively expressed house-keeping control gene. mRNA quantification analyses were conducted as described [37].

#### 4.3.5. Analysis of X Chromosome Inactivation (XCI)

Analysis of XCI was performed using enzymatic digestion of the unmethylated allele (active allele) and PCR amplification of the methylated allele (inactive allele) of the human androgen receptor gene (*HUMARA*) as previously described [38]. This approach allows analysis of XCI alterations due to X-linked pathogenic variants [39,40]. DNA fragment analysis of PCR products was performed on 3130xl Genetic Analyzer (Applied Biosystems, Waltham, MA, USA). The PCR products were also analyzed using DNA-chip microfluidic electrophoresis on a 2100 Bioanalyzer (Agilent Technologies, Santa Clara, CA, USA). Results were analyzed with “Peak Scanner Software v1.0” (Thermo Fisher Scientific, Waltham, MA, USA). Each digested sample data were normalized with undigested sample values to obtain the % XCI and results were expressed as the ratio of the preferentially inactivated allele, according to the protocol of Torres et al. [40].

### 4.4. Western Blot Analysis of NDUFB11 and Other MRC Complexes Subunits

Skeletal muscle and heart tissues were resuspended in lysis buffer (1% Tx-100, 10 mM Tris-HCI pH 7.6, 150 mM NaCl, 1 mM EDTA) with protease inhibitor and phosphatases (Roche Diagnostic) and then were centrifuged at 11,000× *g* for 15 min (4 °C). Protein concentration was quantified with Micro BCA Kit (Thermo Fisher Scientific, Waltham, MA, USA). Samples were incubated for 5 min in TC 4X buffer (β-Mercaptoethanol) at 95 °C and 30 µg of protein extracts were separated by SDS-PAGE electrophoresis in 12.5% polyacrylamide gels and transferred to polyvinylidene fluoride (PVDF) membranes (GH Healthcare, Chicago, CA, USA). The following antibodies were used for inmunodetection: NDUFB11 (Abcam, 1:1000, Cambridge, UK), NDUFA9 (Abcam, 1:1000), NDUFV1 (Santa Cruz Biotechnology, 1:1000, Dallas, TX, USA), NDUFB8 (Abcam, 1:500), Core2 (Abcam, 1:2000) and COXII (Abcam, 1:1000). SDHA (Abcam, 1:10,000) was used as loading control. Secondary antibodies conjugated to horseradish peroxidase (Cell Signaling Technologies, Danvers, MA, USA) were used to detect the primary antibodies and the reactions were developed with ECL Prime Western Blotting Detection Reagent (GE Healthcare, Amersham, UK) in a ChemiDoc™ MP Imaging System (Bio-Rad, Hercules, CA, USA). The optical densities of the immunoreactive bands were measured using NIH ImageJ software v1.8 (Wayne Rasband, NIH, Bethesda, MD, USA).

### 4.5. Blue-Native PAGE for MRC and Supercomplexes and CI In-Gel Activity (CI-IGA) Assay

Skeletal muscle and heart tissues samples were resuspended in sucrose buffer (440 mM Sucrose, 20 mM MOPS, 1 mM EDTA) with proteases inhibitor (Roche Diagnostics Mannheim, Germany) and then centrifuged at 800× *g* for 5 min (4 °C). Supernatants were collected and centrifuged at 20,000× *g* for 20 min (4 °C). To prepare native proteins, pellets were solubilized in 100 to 200 μL of buffer composed of 1.5 M aminocaproic acid and 50 mM Bis-Tris, pH = 7.0. After optimizing the solubilizing conditions, digitonin was used at a concentration of 4 g/g of protein. Evaluation of the steady-state levels of the MRC complexes and SCs was carried out by blue native electrophoresis (BNE), in one (1D) or two dimensions (2D), and mitochondrial complex I in-gel activity (CI-IGA) assay following previously described protocols [41]. Proteins were transferred to PVDF membranes (GH Healthcare) for immunodetection with the following primary antibodies: NDUFB11 (CI; Abcam, 1:1000), NDUFA9 (CI; Abcam, 1:1000), SDHA (CII; Abcam, 1:10,000), Core2 (CIII; Abcam, 1:2000) and COX5a (CIV; Abcam, 1:1000). Secondary antibodies conjugated to horseradish peroxidase (Cell Signaling Technologies, Danvers, MA, USA) were used to detect the primary antibodies, and the reactions were developed with ECL Prime (GE Healthcare, Amersham, UK) in a ChemiDoc™ MP Imaging System (Bio-Rad, Hercules, CA, USA). Band intensity was measured by ImageJ software v1.8 (Wayne Rasband, NIH, Bethesda, MD, USA).

### 4.6. Statistical Analysis

For statistical analysis, Mann-Whitney U test was used when comparing two independent groups. Data were reported as the mean ± SEM. Values of *p* < 0.05 were considered statistically significant.

## Figures and Tables

**Figure 1 ijms-24-01743-f001:**
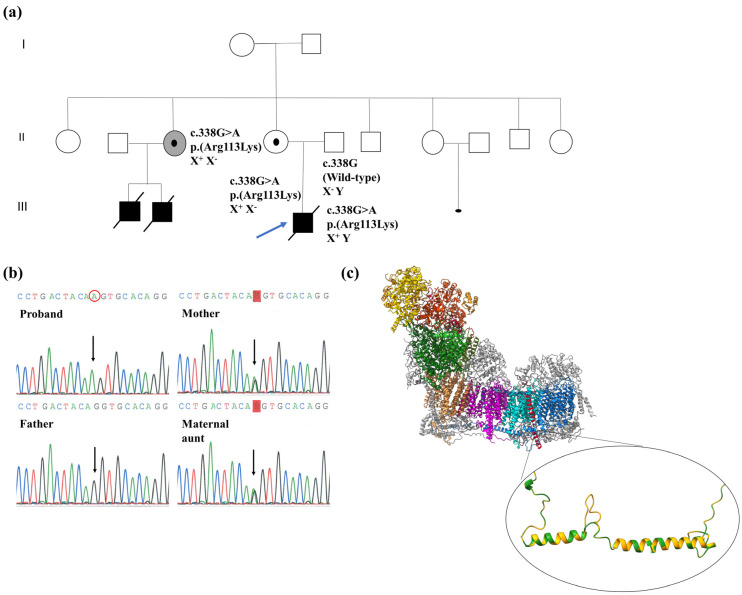
Molecular characterization of the c.338G>A variant in the *NDUFB11* gene. Panel (**a**) Family pedigree: the c.338G>A variant detected in the proband is indicated with a blue arrow. Allelic status of the variant is indicated in the tested relatives. Panel (**b**) Sanger sequencing: electropherograms performed in blood DNA in the proband and relatives. The c.338G>A variant is indicated by an arrow. Panel (**c**) Structural model for c.338G>A genetic variant: the upper panel shows an in silico model of ovine (*Ovis aries*) complex I from The Protein Data Bank (PBD) ID 5LNK [4] displaying the location of the NDUFB11 subunit (colored in red) in the distal module of the CI membrane arm (module P_D-a_) created with ChimeraX; the bottom panel shows the predicted effect of the c.338G>A variant in a human model of the NDUFB11 subunit (PDB ID: 5XTC [19]) representing superposed wild-type (green color) and mutant (yellow color) structure by “SWISS-MODEL” homology modeling and visualized with ChimeraX.

**Figure 2 ijms-24-01743-f002:**
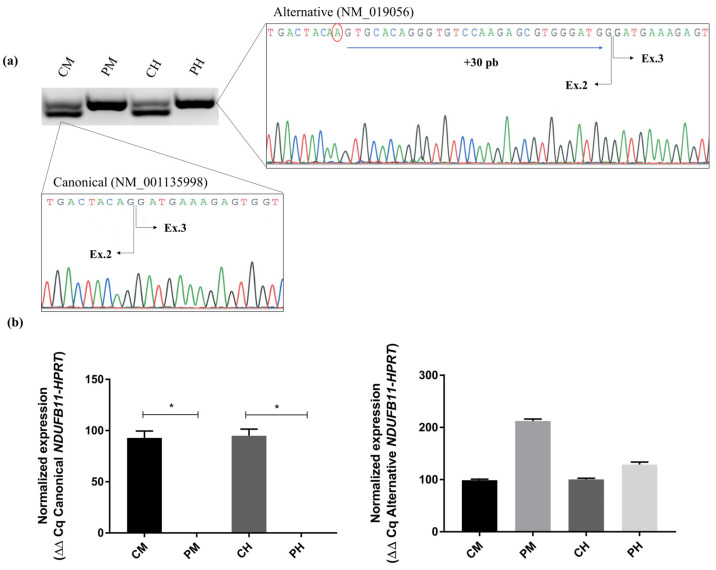
Effects of the c.338G>A variant on the expression of *NDUFB11* transcripts in the proband’s skeletal muscle and heart. Panel (**a**): Electrophoretic and sequencing analysis of *NDUFB11 cDNAs* showing differential expression of the ‘canonical’ (462 bp) and alternative (492 bp) transcripts in the proband’s tissues with respect to controls. Panel (**b**): ‘Canonical’ (left) and ‘alternative’ (right) *NDUFB11* mRNA quantitative expression analysis using reverse transcript-quantitative real-time polymerase chain reaction (RT-qPCR); *NDUFB11* levels were normalized to *HPRT* mRNA levels and expressed as percentage of control (n = 3, mean + SEM). Statistical analysis was performed using Mann-Whitney U test (* *p* < 0.05). CM, control’s skeletal muscle. PM, proband’s skeletal muscle. CH, control’s heart tissue. PH, proband´s heart tissue.

**Figure 3 ijms-24-01743-f003:**
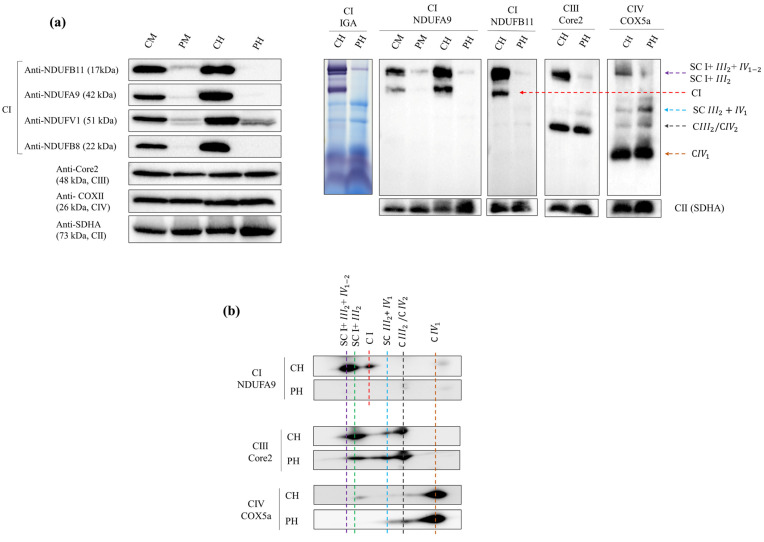
Immunodetection of NDUFB11 and MRC proteins levels and OXPHOS supercomplexes. Panel ((**a**), left) Representative images of immunodetections by Western blot of NDUFB11, NDUFA9, NDUFV1 and NDUFB8 subunits of each CI module (N, Q and P). Core2 and COXII were used for CIII and CIV analysis, respectively. SDHA (CII) was used as loading control. Panel ((**a**), right) in-gel CI activity (CI-IGA) and 1D-BNE showing OXPHOS complexes I to IV (CI-CIV) and supercomplexes (SCs). Panel (**b**) 2D-BN/SDS-PAGE of the steady-state levels of respirasome, SC I + III_2_, SC III_2_ + IV_1_, CIII_2_, CIV_2_, CIV_1_. Respirasome or SC I + III_2_ + IV_1-2_: supercomplex containing CI, CIII_2_ and CIV_2_ or CIV_1_. SC I + III_2_: supercomplex containing CI and CIII_2_. SC III_2_ + IV_1_: supercomplex containing CIII_2_ and CIV_1_. CIII_2_: dimeric CIII. CIV_2_ dimeric CIV. CIV_1_: monomeric CIV. CM, control’s skeletal muscle. PM, patient’s skeletal muscle. CH, control’s heart tissue. PH, patient´s heart tissue.

**Table 1 ijms-24-01743-t001:** Mitochondrial respiratory chain enzyme activities in proband’s skeletal muscle.

MRC Complex	Patient ^1^	Controls ^2^
CI (NADH-Decylubiquinone oxidoreductase)	3.4	15–64
CII (Succinate deshydrogenase)	29	26–65
CIII (Decylubiquinol-cytochrome c oxidoreductase)	126	40–89
CIV (Cytochrome c oxidase)	90	70–228
I + III (NADH cytochrome c reductase)	3.2	8–72
II + III (Succinate cytochrome c reductase)	14	11–25
CS (Citrate synthase)	268	105–350

^1^ Activities of the MRC complexes are expressed as percentage relative of CS specific activity. ^2^ 2.5th–97.5th percentile control range (n, 110).

## Data Availability

The databases and blots analyzed during the current study are available from the corresponding author upon reasonable request.

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
