# Peer review of "A Novel Mutation Associated with Neonatal Lethal Cardiomyopathy Leads to an Alternative Transcript Expression in the X-Linked Complex I NDUFB11 Gene"

_ijms, 2023, doi:10.3390/ijms24021743_

Round 1

Reviewer 1 Report

Manuscript # 2055091

In the current manuscript titled “  A novel mutation associated with neonatal lethal cardiomyopathy leads to an alternative transcript expression in the X-linked complex I NDUFB11 gene” The authors have identified the novel mutation in X-linked complex I NDUFB11 gene, a nuclear gene encoding the structural subunit of mitochondrial complex I. The authors have found that the NDUFB11 gene has two variants including a canonical “short” transcript and an alternative “long” transcript. Mutation in short transcript leads to the improper synthesis of NDUFB11 protein synthesis which causes alteration in mitochondrial electron transport chain complex (NADH:  Ubiquinone oxidoreductase subunit B11 of the mitochondrial respiratory chain (MRC) complex I (CI), which is necessary for oxidative phosphorylation (OXPHOS) and ATP production. The authors have emphasized that mutation in the NDUFB11 gene caused neonatal lethality cardiomyopathy. This is a novel finding and an interesting study. I have the following comments for this study.

Comments:

1)    Overall study design is good and involves all human samples.

2)    In this study, we found that only male samples were analyzed but not females. I  also found that male newborns were dying either 48-55 hours of birth whereas female newborns were dying 6 months- and longer. So, I wanted the authors to discuss if this is true and related to the gender-specific early and severe effects which have clinical relevance. This could be interesting for readers and gain some knowledge about this disease.  

3)    I have concerns that in Figure 3 the authors have not mentioned any MW for the WB image, we are not sure which is the right band size. Please use the MW label.

4)    In Figure 3 WB image, the authors have not used any loading control such as beta-actin or GAPDH, or mitochondrial loading control to show equal loading.

5)    Please use the arrow mark in the WB image for different bands of the mitochondrial complex in Figure 3a.

Author Response

1) Overall study design is good and involves all human samples.

Thank you very much for your comments.

2) In this study, we found that only male samples were analyzed but not females. I also found that male newborns were dying either 48-55 hours of birth whereas female newborns were dying 6 months- and longer. So, I wanted the authors to discuss if this is true and related to the gender-specific early and severe effects which have clinical relevance. This could be interesting for readers and gain some knowledge about this disease.

Thank you very much for your fruitful comment that will improve substantially the discussion section of the manuscript.

We used blood samples for genetic studies from the proband (male), and his relatives: mother, father, and one of the maternal aunts (II-2) who was revealed to have an incidental myocardiopathy. For functional and validation studies we were able to obtain cardiac and muscle samples from the proband. Unfortunately, we cannot be able to get samples from the two male descendants of II-3 who were severely affected having dilated cardiomyopathy and who died during the perinatal period and at 3 months of ages (III-1 and III-2), nor from the III-4 malformed fetus from a voluntary interrupted pregnancy.

X-linked disorders are characterized by more severe clinical symptomatology in male subjects and a higher amount of male patients. This phenomenon can be explained by the fact that males, compared to females, only receive a single copy of the X chromosome. On the other hand, females receive two copies of the X chromosome, and one of these copies is inactivated by the X chromosome inactivation (XCI) in the first stages of embryonic development to compensate for double X chromosome genetic dosage. In this regard, completed or skewed XCI could explain why female heterozygous carriers of pathogenic variants in the NDUFB11 gene are either asymptomatic or show different susceptibility to express phenotypes with respect to very severely affected males harboring hemizygous pathogenic variants. Regarding NDUFB11, it has been previously reported in healthy females: i) two healthy mothers of three females showing an X-linked dominant MLS syndrome with histiocytoid cardiomyopathy, epilepsy, developmental delay, failure to thrive, and hypotony [reference 24 in the manuscript], ii) a healthy mother and a sister of a 23-year-old male displaying sideroblastic anemia, myopathy, and lactic acidosis [ref. 17], and iii) the healthy mother of our proband.  Despite that, it has also been reported an XX abortion at 24 weeks of gestation due to cardiac alterations having NDUFB11 pathogenic variant [ref. 24], as well as, several female patients showing severe cardiac disorders [refs. 15 and 16].

Besides, loss of function variants (p.Trp85*, p.Arg88*, p.Tyr108* and p.Arg134Serfs*3) was the type of mutation present in all reported symptomatic females probands, in contrast with missense variants carrying by severe (p.Glu121Lys) or mild affected (p.Pro110Ser, p.Ser96Pro) males, or with an in-frame variant (p.Phe93del) carried by males with mild sideroblastic anemia [ref. 28]. Interestingly, our early-death male proband carried a splicing loss of function variant. In summary, the manifested phenotype in patients having a pathogenic variant in the NDUFB11 gene is presumably underlaid by a complex interplay between the type of variant, the extent of dysfunctional OXPHOS due to complex I deficiency, and in females by the presence or not of skewed XCI.

We have added several lines in the manuscript discussion section to discuss these arguments.

3)    I have concerns that in Figure 3 the authors have not mentioned any MW for the WB image, we are not sure which is the right band size. Please use the MW label.

MW has been added in Figure 3a. Original blots were sent for revision in the original submission.

4)    In Figure 3 WB image, the authors have not used any loading control such as beta-actin or GAPDH, or mitochondrial loading control to show equal loading.

We agree with the reviewer's comment that beta-actin and GAPDH are common loading controls in WB analysis. However, we decided to use SDHA, one of the four subunits of the mitochondrial complex II, as a mitochondrial loading control to normalize OXPHOS protein levels in WB, because we detected normal activity of complex II in muscle tissue, complex II is not involved in respirasome biogenesis (supercomplex I+III+IV or respirasome), and NDUFB11 is a subunit of complex I. Likewise, SDHA has been previously used as a loading control and normalizer for NDFUB11 WB-analysis [ref. 28 in the manuscript] and other complex I subunits [ref. 3],  and it is also commonly used as a reference to analyze mitochondrial respiratory complexes and supercomplexes assembly. We had already indicated in the Figure 3 legend of the original submission that SDHA was used as a loading control.

5)    Please use the arrow mark in the WB image for different bands of the mitochondrial complex in Figure 3a. 

We added colored arrows to Figure 3a.

Reviewer 2 Report

General comments

Authors showed that a novel exonic c.338G>A variant in the nuclear NDUFB11 gene is a splicing variant. It is associated with mitochondrial X-linked isolated CI deficiency and neonatal lethal hypertrophic cardiomyopathy. Authors suggest that the canonical short transcript of NDUFB11 gene is required for the correct protein synthesis, as well as for CI assembly and activity, while the alternative long NDUFB11 transcript seems to have no functional role.

The study appears to have appropriate methodology. The data are clearly presented. The manuscript is well and clearly written. Few sentences need some stylistic amendments (see below).

In the whole text authors should check and correct the following thing: abbreviation for base pair is bp, not pb.

There are few comments which may be useful:

Abstract

Line 32 – In the sentence ‘Our results support that the canonical ‘short’ transcript is required for the proper NDUFB11 protein synthesis…” something is missing after support. Please correct it.

Discussion

Line 202 – sentence ‘CI is dually encoded by both mtDNA 202 and nDNA…’ is not clear, it might suggest that the same gen can be found in both places (nucleus and mitochondria), please rewrite it.

Line 254-  instead of poor understable please use poorly understable.

Line 271 – in the Discussion section I would advise against referring any table (Table 1 in this case).

Line 311 – instead of ‘transcript seems have’ please write ‘transcript seems to have’.

Author Response

In the whole text authors should check and correct the following thing: abbreviation for base pairis bp, not pb.

There are few comments which may be useful:

Abstract

Line 32 – In the sentence ‘Our results support that the canonical ‘short’ transcript is required for the proper NDUFB11 protein synthesis…” something is missing after support. Please correct it.

Discussion

Line 202 – sentence ‘CI is dually encoded by both mtDNA 202 and nDNA…’ is not clear, it might suggest that the same gen can be found in both places (nucleus and mitochondria), please rewrite it.

Line 254-  instead of poor understable please use poorly understable.

Line 271 – in the Discussion section I would advise against referring any table (Table 1 in this case).

Line 311 – instead of ‘transcript seems have’ please write ‘transcript seems to have’.

Thank you very much for your consideration and your stylistic and spelling comments.  We have checked your suggestions and made the necessary changes throughout the manuscript.

Reviewer 3 Report

By this paper, authors detected a novel splicing variant  of the NDUFB11 gene associated with fatal HCM and characterized it from a biochemical point of view.

Introduction is clear and nicely introduce the topic and conclusion is supported by results. Overall this paper has a good scientific soundness with no need of revision.

Author Response

Thank you very much for your consideration and comments on our manuscript.